# Analysis of Viscoelastic Damping Effect on the Underwater Acoustic Radiation of a Ring-Stiffened Conical Shell

Zhanyang Chen [1] , Qingtao Gong [2],*, Weidong Zhao [1] and Hongbin Gui [1]

1 School of Ocean Engineering, Harbin Institute of Technology at Weihai, Weihai 264209, China; chenzhanyang@hit.edu.cn (Z.C.); zwd-ship@hotmail.com (W.Z.); guihongbin@sina.com (H.G.)
2 Ulshan Ship and Ocean College, Ludong Universtity, Yantai 264025, China
* Correspondence: gongqt@ldu.edu.cn

**Abstract:** Hydroacoustic radiation from submarine power compartments has been the focus of research for many years. In this work, a ring-stiffened conical shell with bases is employed as the target model. The self-factors of viscoelastic damping, such as the damping thickness and damping location, are selected to study their effect on the underwater acoustic radiation, which will give a better understanding of the optimal scheme for applying damping. Firstly, based on a combination of finite element method (FEM) and boundary element method (BEM), the underwater vibration and acoustic radiation of the whole structure are analyzed. Secondly, taking a clamped-free sandwich beam as an example, through comparison with the published numerical data, the present damping application method and mesh density are verified. Finally, the influence of viscoelastic damping parameters on acoustic radiation is studied. The results show that the viscoelastic damping thickness has a significant influence on the amplitude of radiated noise but does not change the distribution of the acoustic field. Furthermore, the damping on the web may achieve the best vibration damping effect in this work.

**Keywords:** conical shell; vibration; acoustic-structural mode; underwater acoustic radiation; viscoelastic damping

## 1. Introduction

Underwater noise is an important factor restricting the performance of submarines and ships [1]. Since the power cabin is located at the stern of a submarine, the vibroacoustic characteristics of the submarine stern directly determine the vibroacoustic level of the whole hull [2,3]. Therefore, research on the vibroacoustic characteristics of the submarine stern structure has great significance for the performance of submarine. For the vibroacoustic simulation of a submarine, the middle part of the submarine can be simulated as a multi-cabin cylindrical shell, and the stern structure can be simulated as a ring-stiffened conical shell. The vibroacoustic problem of the regularly shaped structures has been thoroughly studied [4,5], while there are few studies on the effect of viscoelastic damping on underwater acoustic radiation in the open literature [6]. Thus, the suppression of the vibration and radiation from underwater structures using viscoelastic damping layers is urgent and necessary.

Numerous studies on vibration characteristics of shells in the air have been carried out in the past two decades [7–10]. However, the structural vibration in the fluid medium is quite different from that of air or vacuum. In addition to the vibration of structure, there is an interaction between the structure and fluid medium. When the structure is excited to produce vibrations, the vibration boundary may compress the surrounding fluid to make the vibration propagate outwards in the form of a wave, i.e., acoustic radiation. Meanwhile, the vibration of the surrounding fluid will occur and act on the structure as a radiation force, which is called the acoustic–structure coupling problem [11,12]. The underwater

acoustic radiation of the submarine occurs in a complex environment with an interaction of the structural vibration and water medium. As for the submersed equipment, applying damping on the structure is an effective approach to reduce vibration and noise. Since the viscoelastic materials have high damping characteristics, researchers have carried out numerous studies in this field, including on the free damping beam, plates, shells and influence of external conditions [13–19]. However, the effect of self-factors of damping, such as the damping thickness and damping location, on the acoustic radiation field has not been thoroughly studied. Therefore, it is essential to study the effect of viscoelastic damping on underwater vibration and acoustic radiation.

Generally speaking, for the underwater acoustic radiation of elastic structures, the FEM and BEM are two basic numerical methods. The FEM is a traditional but effective way that requires the discretization of both structure and fluid [20,21]. Furthermore, for large-scale complex structures or far-field radiation problems, the calculation cost will be extremely large. Therefore, the FEM is applied to obtain the vibration responses of the fluid–structure interface [22], and it is used for the low-frequency acoustic and near-field problems [23,24].

Differing from FEM, BEM is governed by the Helmholtz equation, and can satisfy the boundary condition at infinity and simplify the calculation. Researchers have analyzed the vibroacoustic responses based on the BEM method [25–27]. However, BEM requires a large number of computing resources to store a large amount of data. Hence, the combination method of FEM and BEM is usually applied for fluid–structure coupled vibration problems employing the BEM modeling the fluid and the FEM modeling the structure. Since only the structural surface needs to be discretized in the BEM method, the calculation cost would be greatly reduced. Researchers have confirmed that the combination method is a direct and feasible way [28–32].

It is worth mentioning that underwater sensor networks (UWSNs) are considered powerful technology to observe and explore underwater environments. The Medium Access Control (MAC) layer protocol is the most important part. Since certain physical restrictions and distinctive features of underwater environments can be considered during the development of MAC, UWSNs has been a valuable research direction in approaching the impact of underwater acoustic characteristics [33]. Han et al. (2014) analyzed the impacts of node deployment strategies on localization performances in a 3D environment. To achieve this goal, three different deployment schemes in 3D UASNs were presented and compared [34]. As for the routing design of UASNs in uncertain ocean environments, Chen et al. (2021) presented the AAD-FPVR routing algorithm. The proposed algorithm considers node drifting information, uncertain ocean ambient noise [35].

Underwater acoustic radiation has always been our research focus. The purpose of the work is to determine the optimal scheme for applying damping. To achieve this goal, coupling between the ring-stiffened conical shell and fluid medium is simulated by using the FEM, and the underwater acoustic radiation field is predicted by using the BEM. By analyzing and quantifying the damping parameters in different conditions, the effect of viscoelastic damping on the underwater acoustic radiation is obtained, which provides references for the acoustic stealth performance of submarine.

## 2. Method of Fluid–Structure Coupling Vibration and Acoustic Radiation

The vibration of the structure in the medium is different from that of vacuum or air. In addition to the vibration of the structure itself, it also interacts with the surrounding medium. In general, the analytical method is suitable for simple structures, and the vibration and acoustic radiation of most elastic structures can only be solved by a numerical method whose fundamental equation is the Helmholtz equation.

The sound pressure *p(x, y, z)* should satisfy the Helmholtz equation as follows:

$$\nabla^2 p(x,y,z) + k^2 p(x,y,z) = 0. \ (x,y,z) \in \Omega \tag{1}$$

where $\nabla$ denotes the Laplace operator. $\nabla^2 = \frac{\partial^2}{\partial x^2} + \frac{\partial^2}{\partial y^2} + \frac{\partial^2}{\partial z^2}$. $\Omega$ denotes the flow field computational domain. $k$ is the number of waves per second, $m^{-1}$. $k = \omega/c = 2\pi f/c$. $\omega$ is the angular frequency, rad/s. $\omega = 2\pi f$. $f$ is the frequency, Hz. $c$ is sound velocity, m/s.

In this work, the Neumann boundary condition is applied. Since the normal derivative of sound pressure on boundary is known, Equation (1) can be written [6]:

$$\nabla^2 p(x,y,z) + k^2 p(x,y,z) = -j\rho_0 \omega q(x,y,z) \ (x,y,z) \in S \tag{2}$$

where $\rho_0$ is the density of medium. $q(x, y, z)$ is the normal velocity. $j$ is the imaginary unit, $j = \sqrt{-1}$. $S$ is the boundary of $\Omega$.

### 2.1. FEM

The acoustic FEM is mostly applied to analyze enclosure acoustic characteristics. Generally speaking, the acoustic field is discretized into elements, and the acoustic field within the element is determined by the sound pressure on the nodes. The sound pressure distribution of the acoustic field can be obtained by calculating sound pressure on nodes.

It is assumed that the sound pressure $p(x, y, z)$ satisfies Equations (1) and (2) in the bounded domain. Equation (2) is multiplied by the weight function $\widetilde{p}$ and integrated on $\Omega$. The integral expression in the acoustic field will be obtained using integration by parts:

$$\int_\Omega \widetilde{p}\Big(\nabla^2 p(x,y,z) + k^2 p(x,y,z) + j\rho_0 \omega q(x,y,z)\Big) dV = 0 \tag{3}$$

where $\widetilde{p}$ is the weight function.

After a series of derivations, the acoustic dynamic equation can be expressed [6]:

$$\Big([K] + j\omega[C] - \omega^2[M]\Big) \cdot \{p_i\} = \{Q_i\} \tag{4}$$

where $[K]$ is the stiffness matrix. $[M]$ is the mass matrix. $[C]$ is the damping matrix. $\{Q_i\}$ is the acoustic excitation.

The sound pressure in physical space can be obtained by solving Equation (4).

### 2.2. BEM

The BEM does not need to divide the whole acoustic field into the grid, but the surface of the acoustic field needs to be meshed. Thus, the surface mesh is generally required in the BEM. The BEM can be divided into the direct BEM and indirect BEM. The direct BEM requires the mesh to be closed, while the mesh of indirect BEM cannot be closed. Therefore, for the direct BEM, the acoustic fields inside and outside the mesh can be calculated, while the two acoustic fields cannot be analyzed at the same time. The indirect BEM can simultaneously calculate the internal and external acoustic fields.

## 3. Coupled Modal Analysis of Ring-Stiffened Conical Shell with Bases

### 3.1. Establishment of Structural Model

A ring-stiffened conical shell with bases is built in Ref. [36], as shown in Figure 1. The full length is 3200 mm, and the thickness is 7 mm. The diameters of the front-end and back-end are 1750 mm and 840 mm, respectively. The rib interval is defined as 150 mm and the rib thickness is 5 mm. The overall material is steel. The relevant parameters, i.e., elastic modulus, Poisson's ratio, density and loss factor, are defined as 210 GPa, 0.3, 7800 kg/m$^3$ and 0.001, respectively. The bases are mainly composed of plates, webs and ribbed slabs. The plate thickness is 10 mm, and the thicknesses of web and ribbed slabs are all 5 mm. The length of the base is 1200 mm. The width of the web is 100 mm, and the rib interval is 150 mm. The distances of a plate's front-end and back-end to shell surface are 370 mm and 532.5 mm, respectively. The distance from web to the longitudinal center plane of shell is 325 mm.

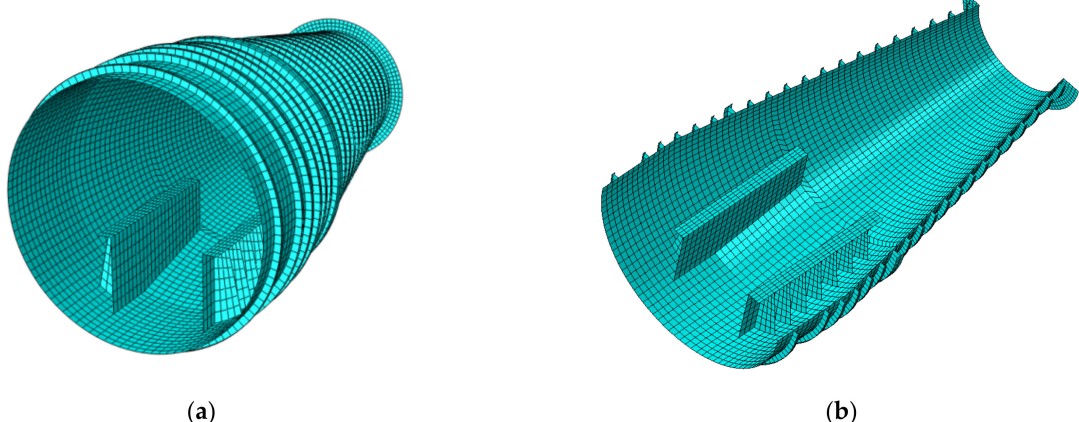

(**a**)          (**b**)

**Figure 1.** Finite element model of the ring-stiffened conical shell with bases: (**a**) axonometric drawing; (**b**) cutaway view.

In this paper, a combination of FEM and BEM is applied to deal with the underwater vibration and acoustic radiation of the ring-stiffened conical shell. Since the coupling between the conical shell and water is analyzed by applying the FEM in this section, the finite element model of the fluid medium is established in Figure 2. The conical shell in Figure 2 is modeled by the structural finite element quadrilateral meshes, and the water is modeled by the acoustic finite element tetrahedral meshes. To achieve a compromise between computation efficiency and accuracy, the size of the finite element model of water is 10 m long × 10 m wide × 10 m high, and each side is divided into 30 tetrahedral elements. The mesh size of the conical shell is 50 mm × 50 mm and the mesh size of the base is 50 mm × 25 mm. The fluid density $\rho_0 = 1000$ kg/m$^3$. The sound velocity $c = 1500$ m/s. Since the whole conical shell is immersed in the water, the coupling effect outside the conical shell is mainly caused by external flow field, while the inside of the conical shell is considered as a vacuum.

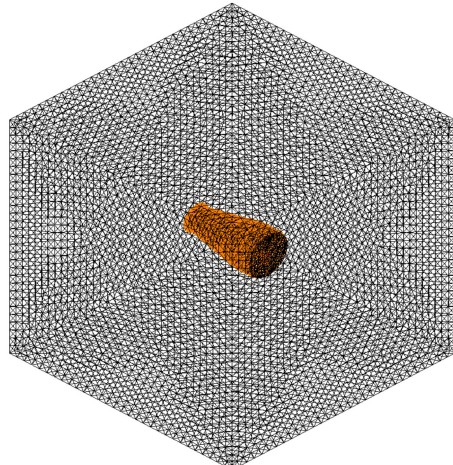

**Figure 2.** Fluid–structure coupled finite element model.

*3.2. Coupled Modal Analysis*

3.2.1. Acoustic–Structure Coupled Modal Analysis

Prior to the acoustic radiation analysis, the acoustic finite element mesh is introduced and coupled with structural mesh. The acoustic–structure coupling modal analysis of the ring-stiffened conical shell system is carried out. The first eight-order coupled modes of the ring-stiffened conical shell are presented in Figure 3. As shown in Figure 3, compared with modes of the shell in vacuum, an obvious influence of the fluid medium can be seen. The first two-order vibrations still occur on the shell. However, compared with the vibration in

vacuum, the vibration amplitudes of the shell and bases decreases greatly. The third-order and fourth-order vibrations are presented as the combined vibration of the bases and shell, and the amplitudes of them are basically the same. The fifth-order and sixth-order vibrations are mainly the base vibration, while the seventh-order is mainly the vibration on the shell. This is quite different from the modal shape in vacuum. It can be seen that the vibration of the whole structure is greatly affected by the added mass of water medium. Both the natural frequency and the modal shapes are greatly changed and the amplitudes of vibration are significantly decreased.

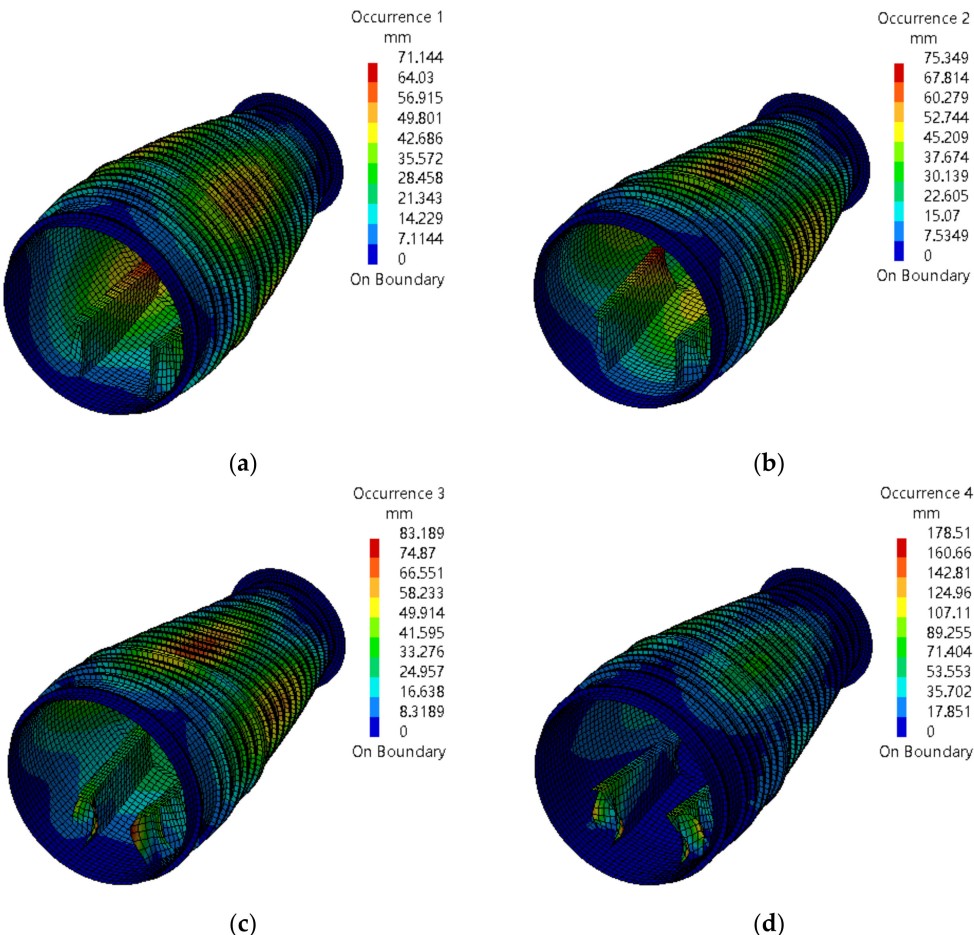

**Figure 3.** *Cont.*

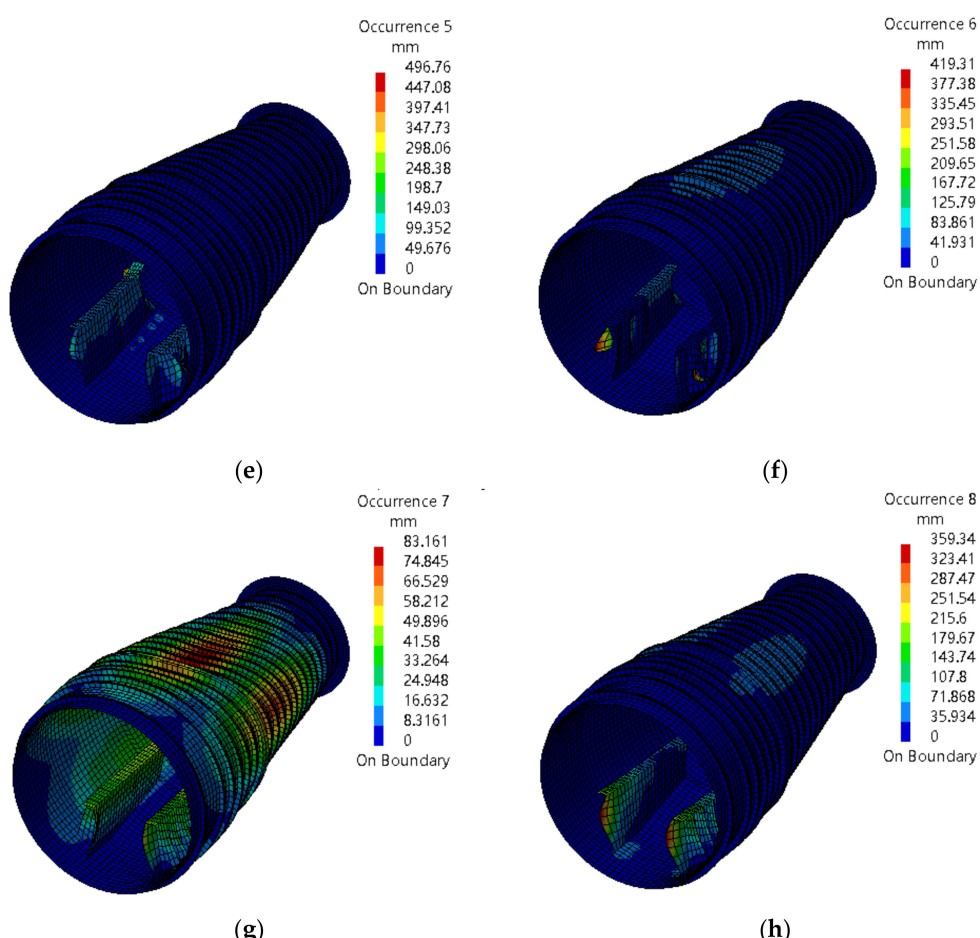

**Figure 3.** Coupled modal shape of the ring-stiffened conical shell in water: (**a**) 1st order; (**b**) 2nd order; (**c**) 3rd order; (**d**) 4th order; (**e**) 5th order; (**f**) 6th order; (**g**) 7th order; (**h**) 8th order.

The natural frequency in a different medium is shown in Table 1. Significant differences are found between the natural frequency in vacuum and in water. Compared with the natural frequency in vacuum, the natural frequency in water decreases significantly, and each order natural frequency decreases by around 100 Hz.

**Table 1.** Natural frequency in different medium/Hz.

| Order | 1 | 2 | 3 | 4 | 5 | 6 | 7 | 8 | 9 | 10 |
|---|---|---|---|---|---|---|---|---|---|---|
| Vacuum | 143.1 | 144.1 | 163.7 | 170.6 | 184.1 | 199.2 | 206.0 | 207.9 | 225.5 | 226.0 |
| Water | 55.4 | 55.6 | 76.0 | 81.1 | 86.1 | 86.6 | 90.0 | 92.5 | 115.6 | 121.4 |
| **Order** | **11** | **12** | **13** | **14** | **15** | **16** | **17** | **18** | **19** | **20** |
| Vacuum | 248.0 | 249.0 | 249.2 | 259.0 | 261.8 | 262.3 | 266.3 | 266.4 | 267.3 | 268.0 |
| Water | 128.1 | 130.7 | 135.5 | 136.8 | 138.8 | 141.3 | 144.4 | 144.6 | 150.9 | 151.4 |

### 3.2.2. Frequency Response of the Ring-Stiffened Conical Shell

To simulate the vibration of the actual engine, the sinusoidal load of 100 Pa is imposed normally on the plates of bases. The vibration responses of the shell in water are analyzed. The procedures are as follows:

- The modal analysis of the structure is carried out firstly;
- The corresponding node information of the outer surface is obtained and imported into acoustic mesh;

- The structural mesh and acoustic mesh are coupled by means of BEM for the acoustic–structure coupling analysis;
- The acoustic characteristic quantity of the structure is obtained.

Due to the limitation of mesh and essence of FEM, the frequency range is 50–2950 Hz, which reaches the limitation of the computer's calculation ability. The step is 50 Hz. The three typical points on shell structure are selected to monitor the responses. As shown in Figure 4, Point 1 is located on the front end of the shell, Point 2 is located on the back end of the shell, and Point 3 is located in the middle part.

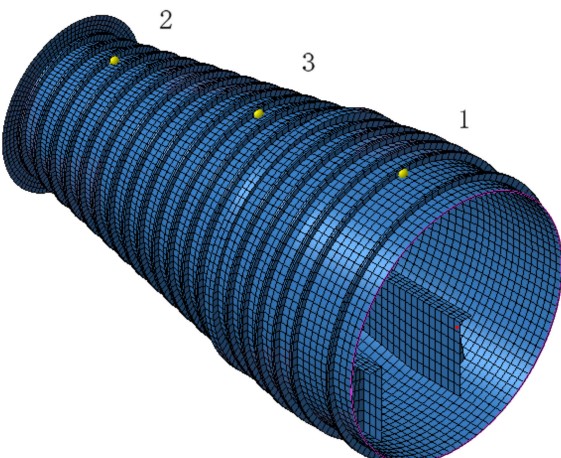

**Figure 4.** Three monitoring points on the shell.

Figure 5 shows the direct vibration displacement responses in vacuum and acoustic–structure coupling responses in water of three monitoring points. It can be seen from Figure 5 that the structural vibration responses have a similar trend in both vacuum and water: the vibration responses of Point 3 are a little larger than those of the other two points. The primary reason is that both ends of the structure are constrained, and the distance from Point 3 to the ends is the greatest, while Point 3 is just above the source. Although Points 1 and 2 are equally distant from the boundaries, Point 1 is closer to the source than Point 2; thus, the vibration responses of Point 1 are larger than those of Point 2. Moreover, since the vibration responses of Point 3 are the largest of them, Point 3 is selected for the comparison, as shown in Figure 6. The values of acoustic–structure coupling responses are lower than those of direct vibration responses. Meanwhile, with the increasing frequency, the acoustic–structure coupling responses decrease continuously, while the direct vibration responses have a relatively stable change. Thus, we can conclude that the acoustic medium has a notable influence on the vibration responses of structure. The acoustic medium not only reduces the whole vibration amplitudes, but also suppresses the high-frequency vibration, which is similar to the damping effect.

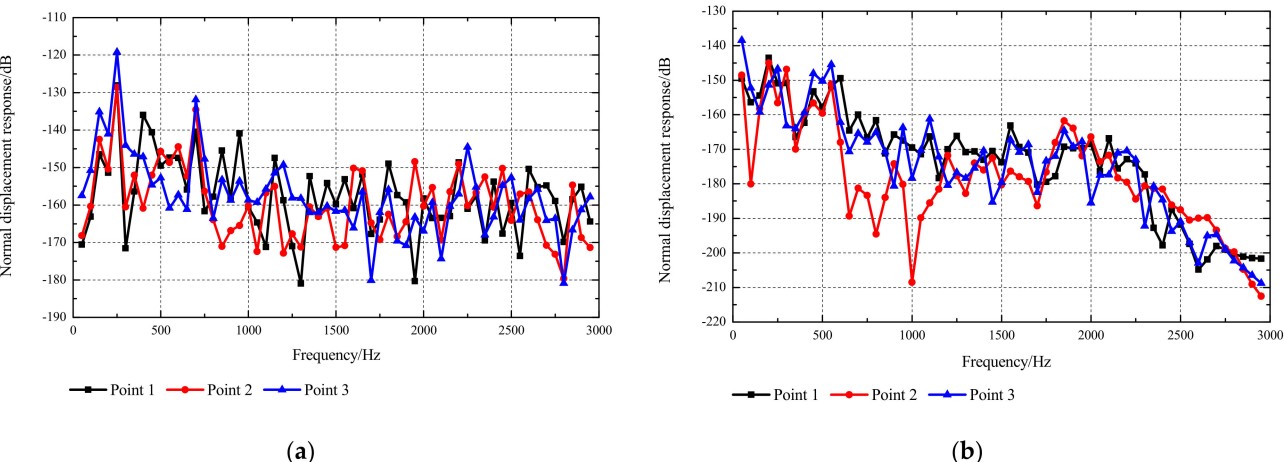

**Figure 5.** Vibration responses of the shell: (**a**) in vacuum; (**b**) in water.

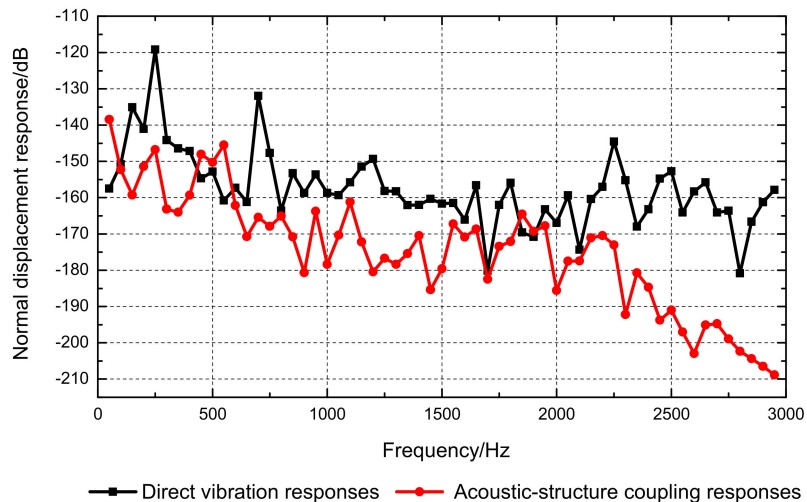

**Figure 6.** Comparison of responses of Point 3.

## 4. Effect of Viscoelastic Damping on Underwater Acoustic Radiation

### 4.1. Numerical Method Validation

Prior to the analysis of underwater acoustic radiation of the ring-stiffened conical shell, natural frequency $f_n$ and loss factor $\eta$ of viscoelastic sandwich beams are simulated and compared with reported numerical results to validate the present method. Figure 7 displays the finite element model of the viscoelastic sandwich beam under clamped-free beam boundary conditions. Since the constitutive model of real viscoelastic materials varies with frequency, the polyvinyl butyral (PVB) is chosen as the viscoelastic damping material in Refs. [37–39]. The material and geometrical properties of the sandwich beam are listed in Table 2.

**Table 2.** Material and geometrical properties of the sandwich beam.

|  | Elastic Faces | Viscoelastic Core | Beam | |
| --- | --- | --- | --- | --- |
| Young's modulus | $6.9 \times 10^{10}$ N/m$^{-2}$ | $1.794 \times 10^6$ N/m$^{-2}$ | Length | 177.8 mm |
| Poisson ratio | 0.3 | 0.3 | | |
| Density | 2766 kg/m$^{-3}$ | 968.1 kg/m$^{-3}$ | Width | 12.7 mm |
| Thickness | 1.524 mm | 0.127 mm | | |

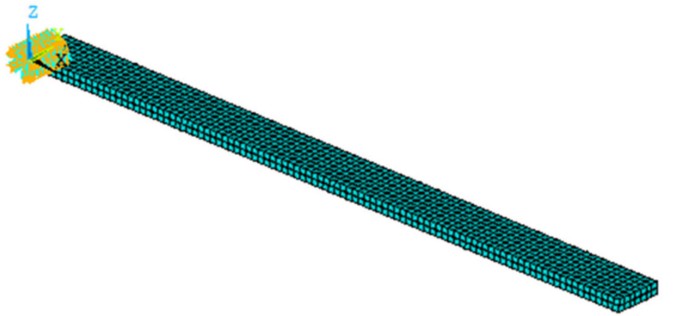

**Figure 7.** Finite element model of the viscoelastic sandwich beam.

The natural frequencies and associated loss factors of the clamped-free sandwich beam corresponding to the first four modes are presented and compared in Table 3. As shown in Table 3, natural frequencies and loss factors are close to the results in Ref. [37]. Therefore, the frequency-dependent viscoelastic model and mesh size can be used in the subsequent simulation of underwater acoustic radiation of the ring-stiffened conical shell.

**Table 3.** Comparison of the results of the sandwich beam.

| Mode | The Present Method | | Ref. [37] | |
|---|---|---|---|---|
| | $f_n$ (Hz) | $\eta$ | $f_n$ (Hz) | $\eta$ |
| 1st | 81.71 | $1.28 \times 10^{-3}$ | 81.79 | $1.37 \times 10^{-3}$ |
| 2nd | 494.14 | $6.20 \times 10^{-3}$ | 504.16 | $5.43 \times 10^{-3}$ |
| 3rd | 1323.4 | $8.06 \times 10^{-3}$ | 1380.34 | $9.38 \times 10^{-3}$ |
| 4th | 2449.8 | $1.19 \times 10^{-2}$ | 2627.87 | $1.36 \times 10^{-2}$ |

*4.2. The Acoustic Field Model and Viscoelastic Damping Material*

After obtaining the vibration response and modes of the submerged shell structure, the acoustic radiation field induced by the structural surface can be calculated. The acoustic fields are established in xoy and yoz planes with 20,000 × 20,000 mm sizes. The intersection of the two fields coincides with the central axis of the conical shell, as shown in Figure 8.

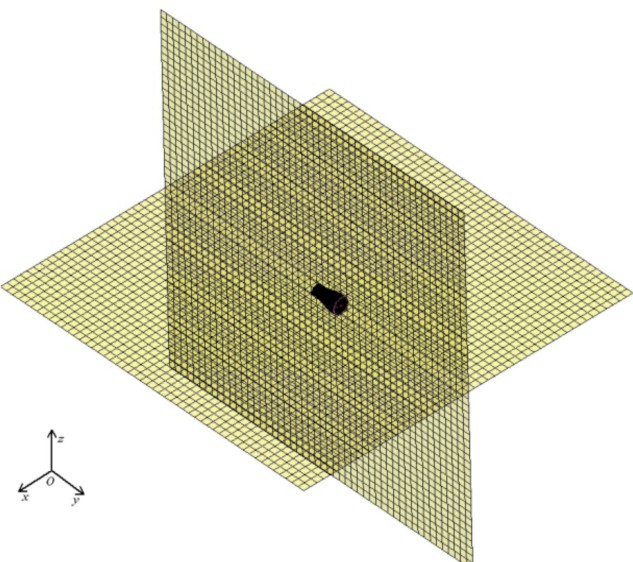

**Figure 8.** Acoustic model of the ring-stiffened conical shell.

As is known, applying damping on the structure can effectively reduce the amplitude of structural vibration and reduce the propagation of noise. However, as for the typical

ring-stiffened conical shell with bases, the effect of geometric parameters and damping location on underwater acoustic radiation remains unclear. Therefore, the acoustic–structure coupling characteristics considering the viscoelastic damping effect are analyzed in this section, which is the key point in this work. The sinusoidal exciting force is imposed normally on the plate of the base with an amplitude of 100 Pa. Thirdly, the BEM is applied for acoustic–structure coupling analysis. The computation frequency is limited to 1500 Hz to avoid the high computational cost.

### 4.3. Damping Thickness

Figures 9 and 10 show the acoustic radiation fields corresponding to damping of 5 mm thick, 10 mm thick and 20 mm thick. The results show that the thickness of the damping layer has little influence on the distribution of sound pressure. However, with the increasing damping layer thickness, the amplitude of the sound pressure level has a marked decline, and the damping effect at a high frequency is better than that at a low frequency. Furthermore, greater thickness is not always better, as a too thick damping layer may increase the mass of the whole structure, which leads to the worse damping effect. Thus, the characteristics of structure vibration and loads should be comprehensively considered in the actual situation.

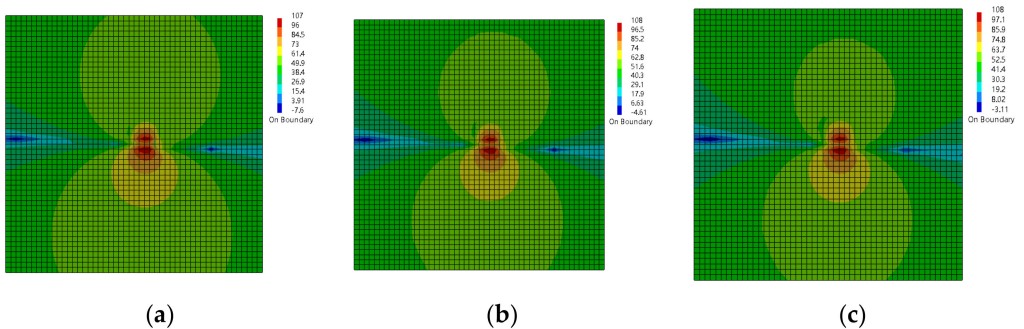

(a)             (b)             (c)

**Figure 9.** Acoustic radiation field corresponding to different damping thickness (50 Hz): (**a**) 5 mm; (**b**) 10 mm; (**c**) 20 mm.

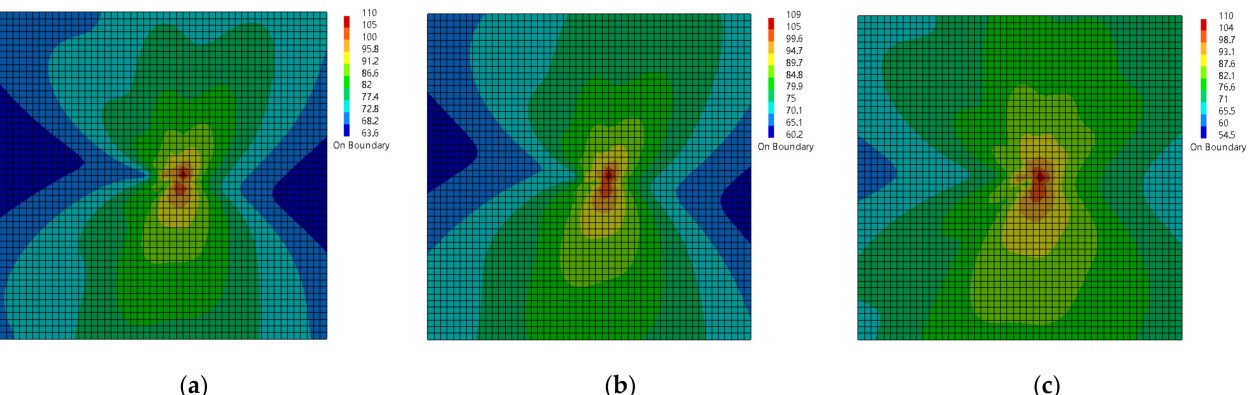

(a)             (b)             (c)

**Figure 10.** Acoustic radiation field corresponding to different damping thickness (1000 Hz): (**a**) 5 mm; (**b**) 10 mm; (**c**) 20 mm.

### 4.4. Location of Damping

Taking the 10 mm thick damping as an example, the damping is laid on the panel, web and ribbed slab respectively, as shown in Figure 11.

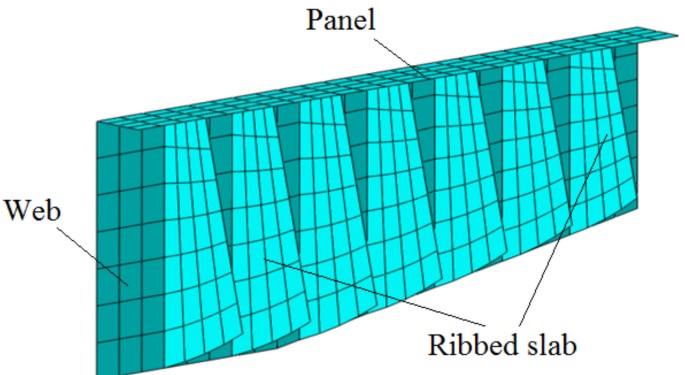

**Figure 11.** Location of the damping on the base.

Figure 12 shows the sound power level corresponding to different damping locations. As shown in Figure 12, the sound power increases at a low frequency after applying damping, while the sound power starts to decrease as the frequency continues to increase. However, the sound power of the undamped structure keeps increasing. By comparing, it is found that applying damping on the structural bases can effectively reduce the sound power level at a high frequency. Moreover, if the damping is laid on the web, the decrease in the sound power level is the greatest and the best vibration damping effect is achieved. If the damping is laid on the ribbed slab, the sound power of the structure would first increase and then decrease, and the peak appears at 800 Hz. If the damping is laid on the plate, the sound power has the same tendency as that of undamped structures, and the overall amplitude is relatively low.

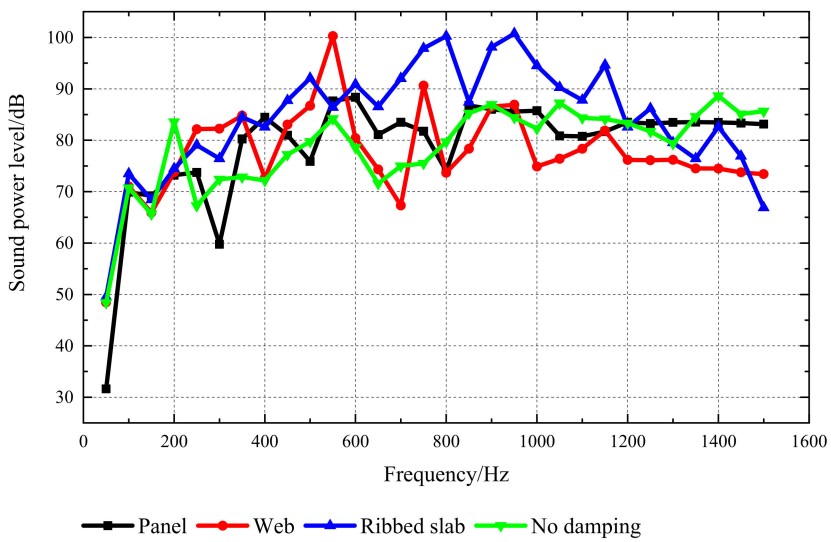

**Figure 12.** Sound power level corresponding to different damping positions.

Figures 13 and 14 present the acoustic radiation field corresponding to the four damping application methods of damping in the yoz plane. As shown, although the damping application method is different, the distributions of sound pressure level are much similar at a low frequency. As the frequency increases, the effect of the damping position on the acoustic radiation field can be observed. Since the interval of natural frequencies is larger at low frequencies, the structures corresponding to the four damping application methods are in the same mode at 50 Hz. The intervals of natural frequencies become small as the frequency increases. When the frequency reaches 1500 Hz, the structure is in different modes, so the acoustic radiation field shows notable difference.

In addition, when the excitation frequency is 50 Hz, although the shapes of the acoustic radiation field are similar, the values of sound pressure are different. The sound pressure level with the damping on the plate is the lowest. Compared with the undamped condition, the maximum value of sound pressure is reduced by 17 dB, while the minimum value decreases by around 9 dB. The changes in sound pressure level with the damping on the web and ribbed slab are almost same. Moreover, it can be seen from Figure 13 that when the damping is laid on the ribbed slab, the sound pressure level in the far-field reaches a minimum. In other words, the radiation range is minimal. It is found that the different locations of damping have different influences on the acoustic radiation field at different excitation frequencies, which has a significant relationship with the vibration modes of structure at the corresponding frequency.

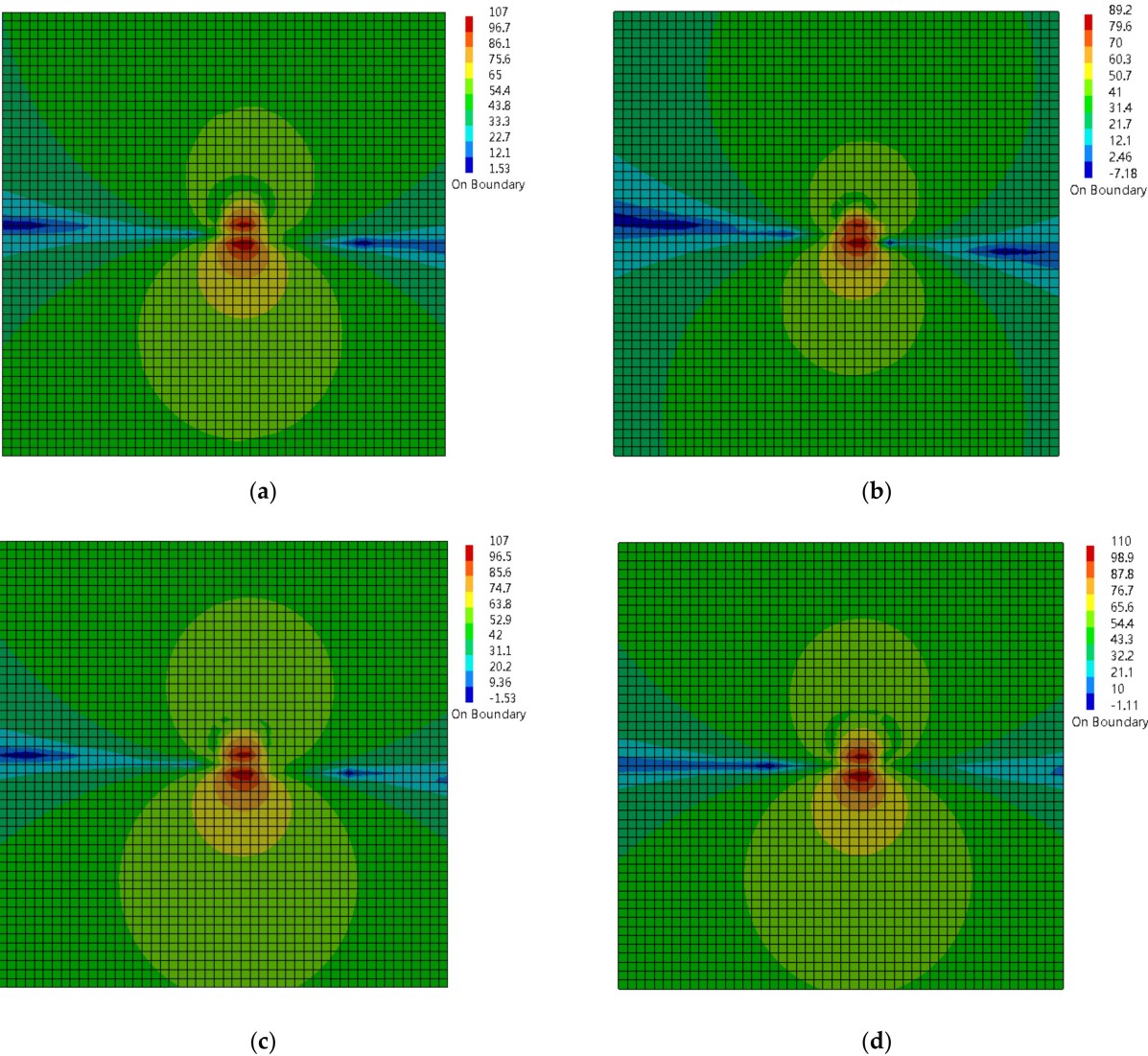

**Figure 13.** Acoustic radiation field in the yoz plane (50 Hz): (**a**) no damping; (**b**) damping on the plate; (**c**) damping on the web; (**d**) damping on the ribbed slab.

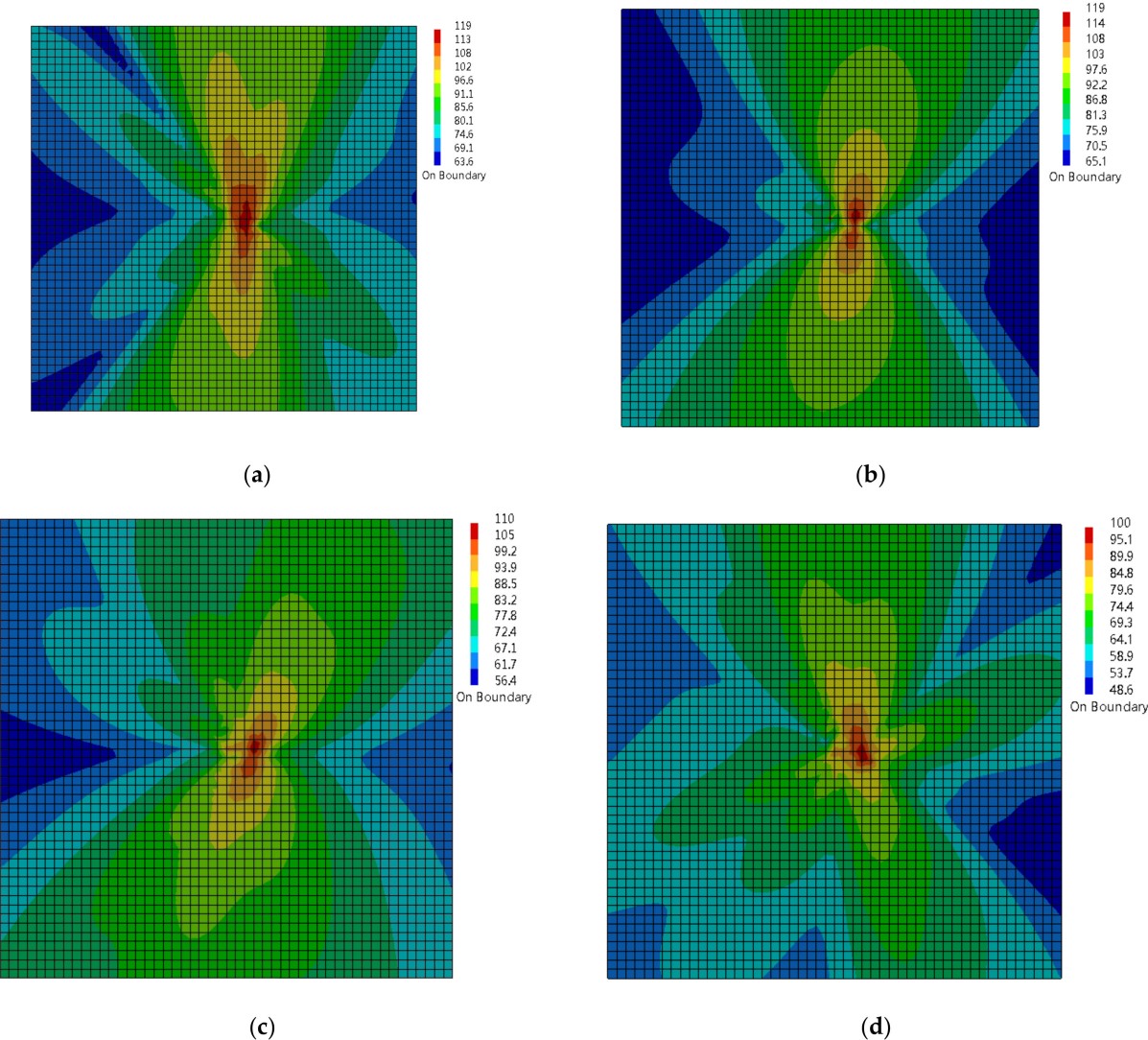

**Figure 14.** Acoustic radiation field in the yoz plane (1500 Hz): (**a**) no damping; (**b**) damping on the plate; (**c**) damping on the web; (**d**) damping on the ribbed slab.

## 5. Conclusions

This paper is concerned with the viscoelastic damping effect on the acoustic radiation field. A numerical model was developed to deal with the vibroacoustic responses of a typical ring-stiffened conical shell by using the combination of FEM and BEM. Some conclusions are drawn as follows:

(1) The water medium can reduce the natural frequency and change the modal shapes. Meanwhile, the effect of suppressing vibration responses at a high frequency is better than that at a low frequency;

(2) The damping effect at a high frequency is better than that at a low frequency. The increase in damping thickness would effectively reduce the amplitude of radiated noise, but does not change the distribution of the acoustic field;

(3) There is a strong correlation between the damping position and the vibration mode of the structure. The damping on the web may achieve the best vibration damping effect, while the damping on the ribbed slab may result in the minimum radiation range.

It is noted that although a series of numerical research is provided in this paper, a detailed validation study would improve confidence in the results. Thus, experimental analysis of vibration and sound radiation in the ring-stiffened conical shell are needed in the future for further validation.

**Author Contributions:** Conceptualization, H.G. and Z.C.; methodology, Z.C.; investigation, Q.G.; software, Z.C.; validation, Z.C.; formal analysis, W.Z.; data curation, W.Z.; writing—original draft preparation, Z.C.; writing—review and editing, W.Z.; supervision, H.G.; project administration, Q.G.; funding acquisition, Q.G. All authors have read and agreed to the published version of the manuscript.

**Funding:** This research was funded by financial support from the financial support from the National Natural Science Foundation of China, grant number U2006229; the Key Research and Development program of Shandong Province, grant number 2019JZZY010125; and the Key Research and Development program of Shandong Province, grant number 2020CXGC010701.

**Institutional Review Board Statement:** Not applicable.

**Informed Consent Statement:** Not applicable.

**Data Availability Statement:** All data, models, or code generated or used during the study are available from the corresponding author by request.

**Conflicts of Interest:** The authors declare no conflict of interest.

## Nomenclature

| | |
|---|---|
| $\nabla$ | Laplace operator, $\nabla^2 = \frac{\partial^2}{\partial x^2} + \frac{\partial^2}{\partial y^2} + \frac{\partial^2}{\partial z^2}$ |
| $\Omega$- | Flow field computational domain |
| $p(x, y, z)$ | Sound pressure |
| $k$ | Number of waves per second (m$^{-1}$), $k = \omega/c = 2\pi f/c$ |
| $\omega$ | Angular frequency (rad/s), $\omega = 2\pi f$ |
| $f$ | Frequency (Hz) |
| $c$ | Sound velocity (m/s) |
| $\rho_0$ | Density of medium |
| $q(x, y, z)$ | Normal velocity |
| $j$ | Imaginary unit, $j = \sqrt{-1}$ |
| $S$ | Boundary of $\Omega$ |
| $V$ | Volume |
| $\widetilde{p}$ | Weight function |
| $[K]$ | Stiffness matrix |
| $[M]$ | Mass matrix |
| $[C]$ | Damping matrix |
| $\{p_i\}$ | Sound pressure matrix |
| $\{Q_i\}$ | Acoustic excitation |
| $f_n$ | Natural frequency |
| $\eta$ | Loss factor |

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
