# Peer review of "Analysis of Viscoelastic Damping Effect on the Underwater Acoustic Radiation of a Ring-Stiffened Conical Shell"

_applsci, doi:10.3390/app12031566_

Round 1
Reviewer 1 Report
Dear authors,
Thanks for having such work. I have left some comments that should be included in the revised version.
Good luck

Reviewer 2 Report
Manuscript ID applsci-1573556 Title: “Analysis of viscoelastic damping effect on the underwater acoustic radiation of a ring-stiffened conical shell” belongs to under water vibroacoustic , more specific to hydroacoustic radiation field.
Concerning the scientific content of the paper
- In the Abstract and in Introduction the aim of the article must be better highlighted, this aim being to determine the optimal scheme for applying damping.
- In paragraph Effect of viscoelastic damping on underwater acoustic radiation, because is missing a paragraph Discussions, it must be better emphasized the viscoelastic damping effects on the acoustic radiation.
- Also, in the same paragraph Effect of viscoelastic damping on underwater acoustic radiation, the obtained results are validated only for the natural frequencies of the clamped–free sandwich beam [32] while the directivity graphs of the acoustic radiation presented in figures 9,10,13,14 are not discussed with respect to the results obtained by other researchers.
- For the previous point 3 I recommend being studied and added at the references ( in order to have similar results) a very important researches in the field published in 2020 namely “Investigation of radiation damping in sandwich structures using finite and boundary element methods and a nonlinear eigensolver” https://doi.org/10.1121/10.0000947 and “Hybrid FEM–SBM solver for structural vibration induced underwater acoustic radiation in shallow marine environment” https://doi.org/10.1016/j.cma.2020.113236.
- The future work is not presented.
Concerning the presentation of the paper with respect to Manuscript Type MDPI journal template
- The References doesn’t respect 100% the form demanded by Manuscript Type MDPI journal template, none of the references have the digital object identifier (DOI) and is missing pp. before the field “pp.” of the mentioned pages ( lines: 323, 325, 325, 327, 329, 331 and all along References).
For the above-mentioned aspects, I advise the minor revision of the paper.
Round 2
Reviewer 1 Report
Dear authors,
A new paragraph requires in the introduction that shows the network and MAC layers and the impact of underwater acoustic features/ characteristics. Some recent references are suggested:
- Han G, Zhang C, Shu L, Rodrigues JJ. Impacts of deployment strategies on localization performance in underwater acoustic sensor networks. IEEE Transactions on Industrial Electronics. 2014 Oct 13;62(3):1725-33.
- Chen Y, Zhu J, Wan L, Fang X, Tong F, Xu X. Routing failure prediction and repairing for AUV-assisted underwater acoustic sensor networks in uncertain ocean environments. Applied Acoustics. 2022 Jan 15;186:108479.
- Alfouzan FA. Energy-efficient collision avoidance MAC protocols for underwater sensor networks: Survey and challenges. Journal of Marine Science and Engineering. 2021 Jul;9(7):741.
Thanks
